# Mixing Enhancement of Non-Newtonian Shear-Thinning Fluid for a Kenics Micromixer

**DOI:** 10.3390/mi12121494

**Published:** 2021-11-30

**Authors:** Abdelkader Mahammedi, Naas Toufik Tayeb, Kwang-Yong Kim, Shakhawat Hossain

**Affiliations:** 1Department of Mechanical Engineering, University of Djelfa, Djelfa 17000, Algeria; abdelkader.mahammedi@gmail.com; 2Gas Turbine Joint Research Team, University of Djelfa, Djelfa 17000, Algeria; toufiknaas@gmail.com; 3Department of Mechanical Engineering, Inha University, 253 Younghyun-dong, Nam-gu, Incheon 402-751, Korea; 4Department of Industrial and Production Engineering, Jashore University of Science and Technology, Jessore 7408, Bangladesh

**Keywords:** Kenics micromixer, numerical simulation, mixing index, non-Newtonian fluids, CMC solutions, low Reynolds number

## Abstract

In this work, a numerical investigation was analyzed to exhibit the mixing behaviors of non-Newtonian shear-thinning fluids in Kenics micromixers. The numerical analysis was performed using the computational fluid dynamic (CFD) tool to solve 3D Navier-Stokes equations with the species transport equations. The efficiency of mixing is estimated by the calculation of the mixing index for different cases of Reynolds number. The geometry of micro Kenics collected with a series of six helical elements twisted 180° and arranged alternately to achieve the higher level of chaotic mixing, inside a pipe with a Y-inlet. Under a wide range of Reynolds numbers between 0.1 to 500 and the carboxymethyl cellulose (CMC) solutions with power-law indices among 1 to 0.49, the micro-Kenics proves high mixing Performances at low and high Reynolds number. Moreover the pressure losses of the shear-thinning fluids for different Reynolds numbers was validated and represented.

## 1. Introduction

Different applications of micromixers can be found in biomedical, environmental industries and chemical analysis, where they are essential components in the micro-total analysis systems for such applications requiring the rapid and complete mixing of species for a variety of tasks [1,2]. Various characteristics of micromixers have been developed to produce fast and homogenous mixing; micromixers are usually classified according to their mixing principles as active or passive devices [3,4,5,6]. Active micromixers need an external energy supply to mix species. Passive micromixers are preferable due to their simple structures and easy manufacturing and greater robustness and stability [7]. To get enhanced mixing flows, the chaotic advection technique is employed as one of the strongest passive mixing methods for non-Newtonian flows. One of the potential chaotic geometries which can present a good method to advance the performances of the hydrodynamics is called the Kenics mixer. A Kenics mixer is a passive mixer created for conditions of laminar flow; it is generally constituted of a series of helical elements, and each element rotated 90° relatives to the second. The helical elements are designed to divide the flow into two or more flows, turn them and afterward recombine them [8,9]. Kurnia [10] observed that the inserted of a twisted tape in a T-junction micromixer creates a chaotic movement that improves the convection mass transfer at the expense of a higher pressure drop. For dean instability, Fellouah et al. [11] investigated experimentally the flow field of power-law and Bingham fluids inside a curved rectangular duct. Pinho and White law [12] studied the effect of dean number on flow behavior of non-Newtonian laminar fluid. For twisted pipes, Stroock et al. [13] realized a twisting flow microsystem with diagonally oriented ridges on the bottom wall in a microchannel. They attained chaotic mixing by alternating velocity fields. Tsai et al. [14] calculated the mixing fluid of non-Newtonian carboxymethyl cellulose (CMC) solutions in three serpentine micromixers. They summarized that the curvature-induced vortices develop in a long way the mixing efficiency. Bahiri et al. [15], using grooves integrated on the bottom wall of a curved surface, studied numerically the mixing of non-Newtonian shear-thinning fluids. They illustrated that the grooves elevated the chaotic advection and augmented the mixing performance. Naas et al. [16,17] and Kouadri et al. [18] used the two-layer crossing channels micromixer to evaluate the mixing rate hydrodynamics and thermal mixing performances, finding that the mixing rate was nearly 99%, at a very low Reynolds number. Kim et al. [19] experimentally characterized the barrier embedded Kenics micromixer. They showed that the mixing rate decreases as the Reynolds number augments for the suggested BEKM chaotic micromixer. Hossain et al. [20] experimentally and numerically analyzed a model of micromixer with TLCCM that can achieve 99% mixing over a series of Reynolds number values (0.2–110).

There are not many works in the literature for non-Newtonian fluid mixing using Kenics micromixers. Therefore, the idea of the current study is to investigate the performance of a microKenics for mixing shear thinning fluids, trying to attain a high mixing quality and pressure drop. Using CFD code, numerical simulations were carried out at Reynolds numbers ranging from 1 to 500 in order to examine the flow structures and the hydrodynamic mixing performances within the concerned Kenics micromixer. Various CMC concentrations were proposed to investigate the chaotic flow formation and thermal mixing performances within the suggested micromixer. In order to get important homogenization of the fluids’ indices and pressure losses will be appraised.

## 2. Governing Equations and Geometry Discretion

Steady conservation equations of incompressible fluid are solved numerically in a laminar regime by using the ANSYS Fluent^TM^ 16 CFD software (Ansys, Canonsburg, PL, USA) [21], which is fundamentally based on the finite volumes method. We choose the SIMPLEC scheme for velocity coupling and pressure. A second-order upwind scheme was nominated to solve the concentration and momentum equations. The numeric’s were ensured and simulated to be converged at 10^−6^ of root mean square residual values. 

A non-Newtonian solution of carboxyméthyl cellulose (CMC) is used as working fluid for the simulation of fluid flows. The density of CMC solutions according to Fellouah et al. [11] and Pinho et al. [12], is 1000 kg/m^3^. The coherence coefficient and the power law indexes of the CMC solutions are indicated in Table 1, where the diffusion coefficient equals 1 × 10^−11^ m^2^/s.

The 3D governing equations for incompressible and steady flows are continuity, momentum and species mass fraction convection diffusion equation:(1)∇U=0
(2)$$\rho U\cdot \nabla U=-\nabla P+\mu \nabla^{2}U$$
(3)$$U\cdot \nabla C=D_{i}\nabla^{2}C_{i}$$
where U (m/s) denotes the fluid velocity, ρ (kg/m^3^) is the fluid density, P (Pa) is the static pressure, μ (w/m∙k) is the viscosity, $C_{i}$ is the local mass fraction of each species by solving the convection-diffusion equation for the *i*-th species. $D_{i}$ is the mass diffusion coeffcient of the species “*i*” in the mixing.

For power-law non-Newtonian fluids the apparent viscosity is:(4)$$\mu_{a}={k\dot{\gamma }}^{n-1}$$
where k (w/m∙k) is called the consistency coefficient and n is the power-law index $\dot{\gamma \;}$(s^−1^) is the shear rate.

For a shear thinning fluid (Ostwald model), the generalized Reynolds number (Re_g_) is defined as [6]:(5)Reg=ρU2−nDhnk8(6n+22)n
where $D_{h}\;$(m) is the hydraulic diameter of the micromixer.

To measure the efficiency of the Kenics micromixers, mixing rate is defined as follows [16]:(6)$$\mathrm{MI}=1-\frac{\sigma }{\sigma_{0}}$$
where σ signifies the standard deviation of mass fraction and characterized as:(7)$$\sigma^{2}=\frac{1}{N}\overset{N}{\underset{i=1}{\sum }}{\left(C_{i}-\bar{C}\right)}^{2}$$

N indicates the number of sampling locations inside the transversal division, $C_{i}$ is the mass fraction at examining point *i*, and $\bar{C}$ is the ideal mixing mass fraction of $C_{i}$, and it is equal to 0.5, $\sigma_{0}\;$(Pa·s^−1^) is the standard deviation at the inlet part.

The boundary conditions are a condition of adhesion on the walls where the velocities are considered to be zero, uniform velocities are executed at the inlets, the mass fraction of the fluid at the inlet 1 equal to 1 and that of the inlet 2 equal to 0, an atmospheric pressure condition is considered at the exit. All walls are considered adiabatic.

The configuration (Figure 1) is based on the Kenics KM static mixer. It consists of a tube with a diameter D = 1.2 mm and a length L = 16.5 mm, with six helical parts. Each part has a thickness t = 0.025 mm and length li = 1.5 mm. The final helical blade element is placed at the distance l = 3 mm from the tube outlet. The angle between the two inlet entrances is α=35°.

## 3. Grid Independence Test

In this work, which investigated fluid mixing in laminar flow in which the convergence is limited by the pressure-velocity coupling, a converged and stable solution was obtained using SIMPLE algorithm. The pressure correction under relaxation factor is given at 0.3, which facilitates the acceleration of convergence for the second order upwind scheme. The convergence of iterative calculations was attained when the specified value of the residual quantities are less than 10^6^. 

To choose an adequate mesh; an unstructured mesh has been generated with tetrahedral elements; several grids were tested for the present proposed geometry (Table 2). Numerical variations of the mixing index are not important after the marked cell size; therefore this can be studied as the better mesh for the calculation. This mesh size with 338,438 cells will give better results with less time as compared to the finer mesh.

## 4. Numerical Validation 

A numerical study of the pressure drop in a T-Junction Passive micromixer to verify the accuracy of the CFD with that of C. Kurnia et al. [10], see Table 3. The comparison illustrated a good agreement where the relative error of the numerical results is less than 1%.

## 5. Results and Discussion

To compare the numerical results a quantitative comparison was made for Newtonian fluids in a Kenics micromixer as shown in Figure 2. The mixing performance of the microKenics was compared with other three micromixers [15]: the SHG micromixer (staggered herringbone), a mixer based on patterns of grooves on the floor of the channel and a 3D serpentine micromixer with repeating “L-shape” units and the TLSCC (two-layer serpentine crossing channels) a micromixer which the principle serpentine channels with an angle of 90° regarding the inlets. 

The mixing indices were compared using the range of Reynolds number between 0.1 to 120. The Kenics micromixer and TLSCC displayed exceptional mixing performance compared to the other two micromixers for Re <30, with superiority of the microKenics, as the Kenics device shows almost complete mixing (MI > 0.999) at low Reynolds numbers (Re = 0.1–5).

The mixing index at the exit of Newtonian and non-Newtonian fluid in Kenics was compared with the curved micromixer of Bahiri et al. [15]. For range numbers of Reynolds (0.1–500) and shear thinning index n (1, 0.85, and 0.6), as shown in Figure 3, the Kenics proved a high mixing efficiency.

The effects of the Reynolds number and the behavior index on the chaotic mixing mechanism were qualitatively analyzed by presenting the contours of the mass fraction at the various transverse planes P1–P7 and the exit. Figure 4 shows the improvement of the mass fraction distribution on the y-plane along the micromixer at Re_g_ = 25 and n = 0.73. The twist of element and the sharp change of angle between blades affected the intensity of the fluid particles’ movement and the mixing performance.

Figure 5 show the streamlines flow in the microKenics for fluid behavior 0.73 and for Re_g_ 0.1 and 50. The flow field enhanced the secondary flow along the micromixer for all cases of Re due to the blade conurbation of the Kenics device.

Figure 6 and Figure 7 present the mass fraction contours for different power-law indices (n = 1 and 0.49), with Reynolds numbers ranging between 0.1 and 50, at the different cross-sectional planes. Table 4 gives the distances between different plans to analyze the local flow behavior. For n = 1, the flow behavior presented by the mass fraction contours shows that the fluid layers for P1 to P4 advance in the same mode of molecular diffusion.

When the Reynolds number increases to 50, the quality of the mixing begins to improve, therefore a homogeneous mixture is obtained at the exit plane in the microKenics for all the values of the behavior index.

Figure 8 shows the variation of mixing index versus generalized Reynolds number for different values of power-law index inside the microKenics. It can be seen that for all values of n, the micromixers have nearly the same high mixing index (Mi = 0.99). 

For low Reynolds numbers (Re = 0.1–5), when the species have more contact time to achieve a perfect mixing with the Kenics, the mixing index loses a part corresponding to nearly 14% of its value. In addition, by increasing the Reynolds number the fluid homonezation augments due to increases of secondary flows and advection, compared with TLCC micromixers for all cases of the power-law index.

Figure 9 shows the evolution of MI along the micro-Kenics, at different planes, with various values of the behavior index and for Re = 1, 5, 10, 25 and 100. For all cases of n, we can see from this figure that MI grows progressively and reaches high values when approaching the exit plane. Thus, as mentioned before we can see in all figures, the mixing performance increases with the increase in the behavior index, for Reg≤50.

The Newtonian fluid with n = 1 is independent of shear rate and maintains a constant value of viscosity among different numbers of Reynolds (Figure 10 and Figure 11). Besides, the decrease of value of n induces the increases of the apparent viscosity of non-Newtonian fluid which also depends on the consistency coefficient of the fluid and the shear rate.

Therefore, Figure 10 indicates that for a known shear rate, where the fluid with a lower power-law index has a higher apparent viscosity furthermore the apparent viscosity increases by reducing the power-law index.

Figure 11 shows the apparent viscosity on line x = 0 at the exit of micro Kenics for all power-law indices. It is obvious that the apparent viscosity decreases by rising the Reynolds number. 

The pressure drop obtained from CFD simulations was compared with the TLCC micromixer [18], for the cases with the same CMC solutions and flow speed. As remarked in Figure 12, the pressure loss of Kenics is less than that from TLCC; so the best advantage has been obtained by the Kenics.

A high mixing performance of the micromixer is generally associated with a high-pressure loss that involves the required energy input for the mixing process. Figure 13 shows the pressure loss increases with the increases of generalized Reynolds and concentration level. It is evident that a decreasing power-law index leads to an increment of the apparent viscosity and consequently a rising pressure loss.

## 6. Conclusions

In this work, mixing of CMC non-Newtonian fluids in a microKenics device was numerically investigated for different regimes (Re = 0.1–500), using CFD code. The analyses showed that the mixing performances of the Kenics micromixers consisting of repeating short twisted helical configurations is better than that of other micromixers at low Reynolds numbers.

It can be achieved that for fluids with all power-law indices studied (n = 0.49, 0.6, 0.73, 0.85, 0.9, and 1) and low Reynolds numbers (less than 8) the micromixer is an excellent one, while for the fluids with Re>12 MI start decreasing for all power-law indices, but for low power law index (n = 0.6), the MI is reached from numbers of Re≥60. At elevated Reynolds numbers (Re≥120), the micromixer performance is improved for all values of the power-law indices. The results confirmed that the apparent viscosity of CMC solutions decreases with the increase of the shear rate, while, the pressure drop increases rapidly with increasing Reynolds number and power-law index. Nevertheless it is still the slightest loss compared to other micromixers in the literature with the same mean flow speed and apparent viscosity.

## Figures and Tables

**Figure 1 micromachines-12-01494-f001:**
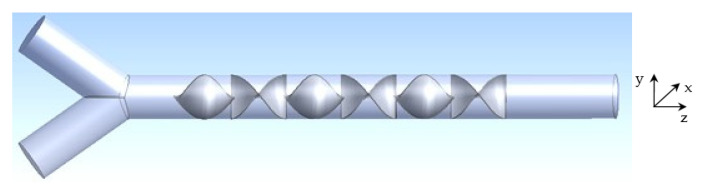
Kenics micromixer.

**Figure 2 micromachines-12-01494-f002:**
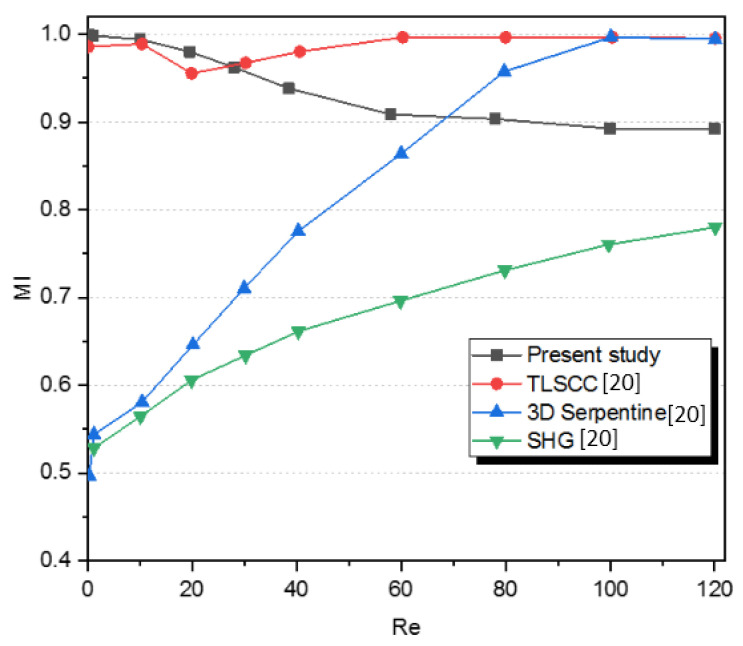
Evolution of mixing index (MI) with Reynolds number for a Newtonian fluid compared with Houssain et al. [20].

**Figure 3 micromachines-12-01494-f003:**
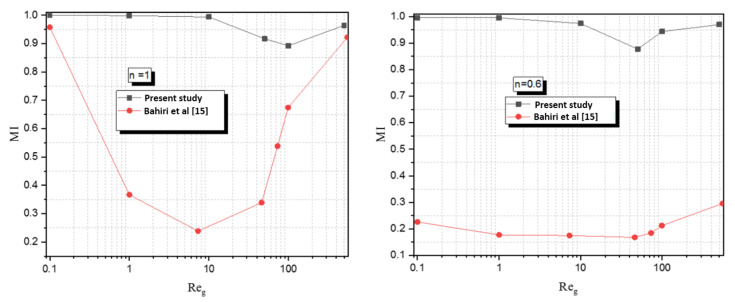
Evolution of mixing index of Reynolds number for a Newtonian and non-Newtonian fluid compared with Bahiri et al. [15].

**Figure 4 micromachines-12-01494-f004:**
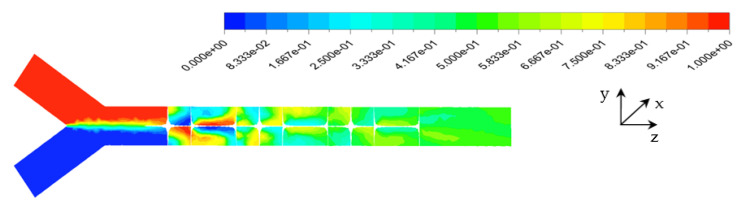
Distribution of mass fraction at Re_g_ = 25, n = 0.73.

**Figure 5 micromachines-12-01494-f005:**
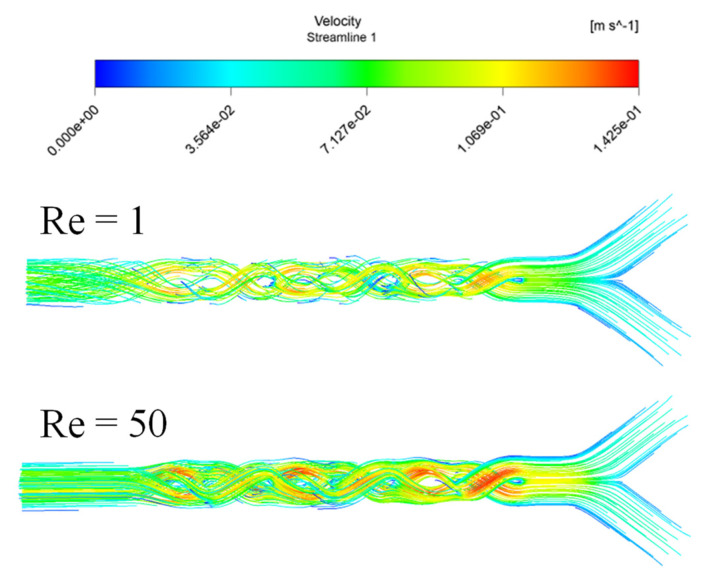
Velocity streamlines of n = 0.73 with different Reynolds number.

**Figure 6 micromachines-12-01494-f006:**
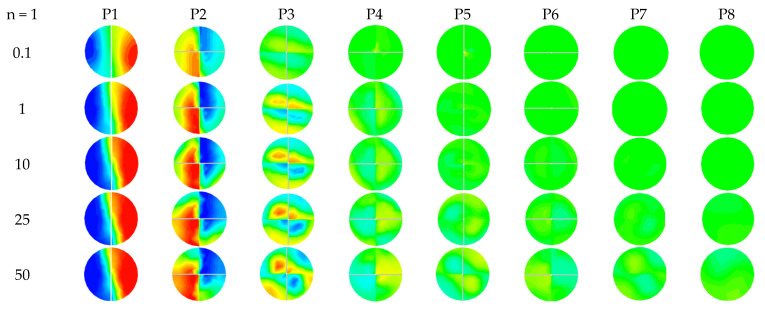
Contour of mass fraction at cross-section P1–P8 for different Reynolds numbers with n = 1.

**Figure 7 micromachines-12-01494-f007:**
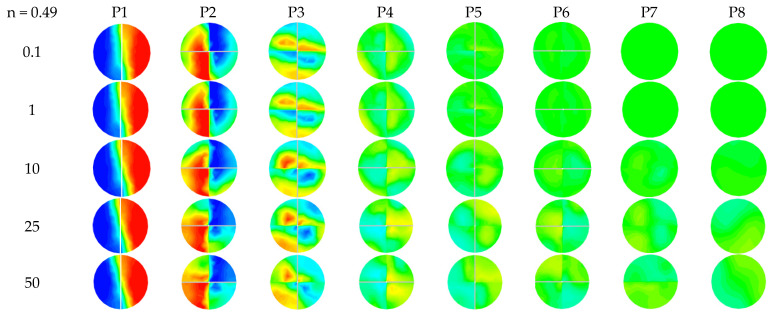
Contours of mass fraction at cross-section P1–P8 for different Reynolds numbers with n = 0.49.

**Figure 8 micromachines-12-01494-f008:**
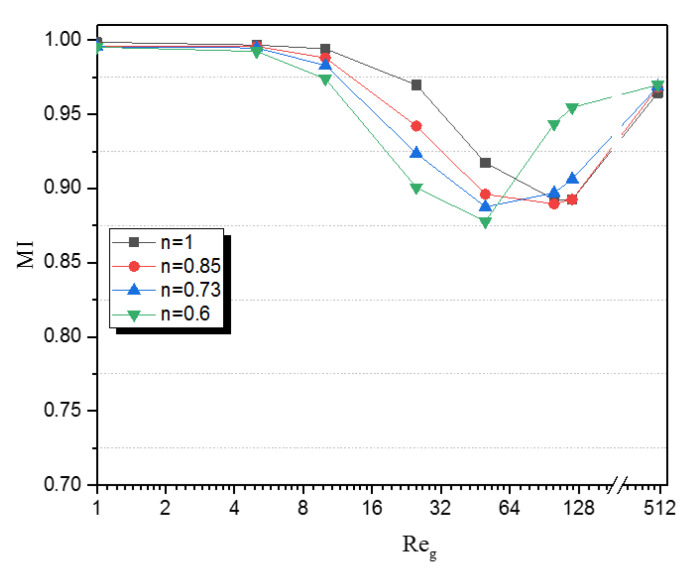
Development of mixing index versus generalized Reynolds number for differences values of power-law index.

**Figure 9 micromachines-12-01494-f009:**
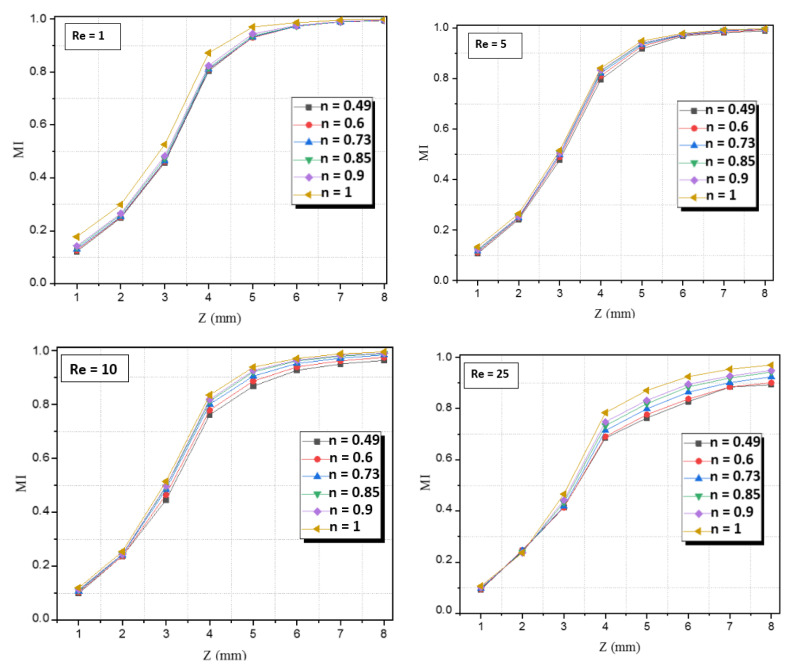
Development of mixing index along the mixing helical for different power-law indices and Reynolds numbers.

**Figure 10 micromachines-12-01494-f010:**
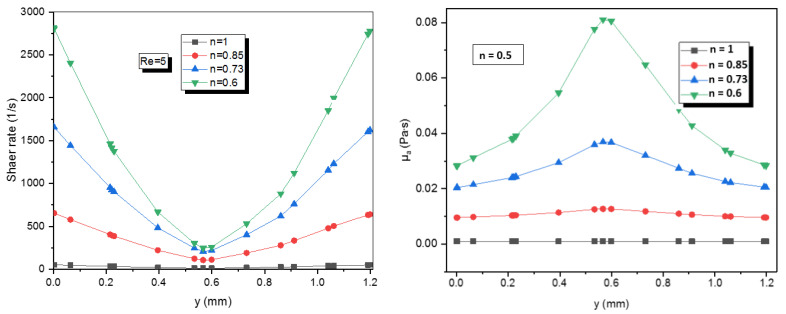
Shear rate profiles and apparent viscosity profiles on line x = 0 at the exit plane for different power-law indices.

**Figure 11 micromachines-12-01494-f011:**
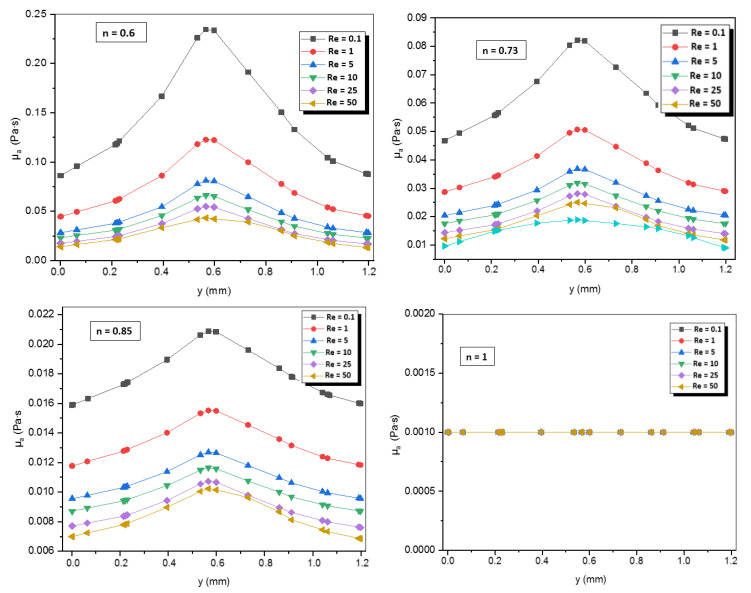
Apparent viscosity profiles on line x = 0 at the exit plane for different Reynolds numbers and different power-law indices.

**Figure 12 micromachines-12-01494-f012:**
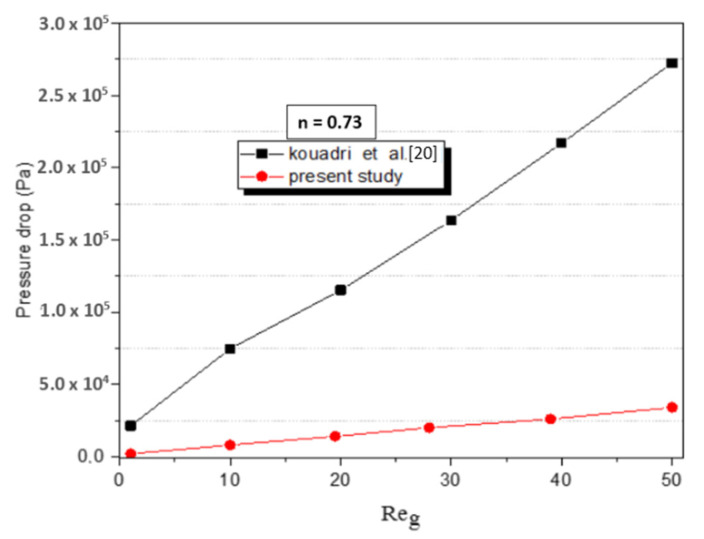
Pressure drop vs. Reynolds numbers compared with TLCC micromixer for n = 0.73.

**Figure 13 micromachines-12-01494-f013:**
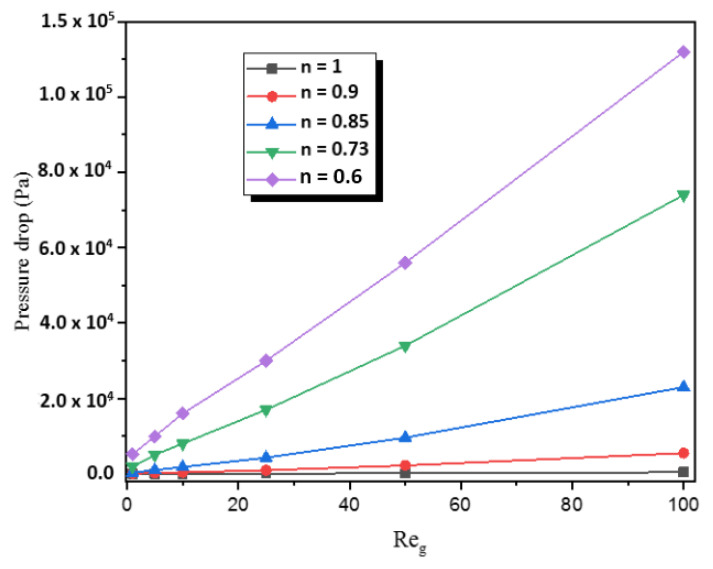
Variations of the pressure drop with generalized Reynolds number for different power-law indices.

**Table 1 micromachines-12-01494-t001:** Rheological properties of CMC solutions.

CMC%	N	$k\;(Pa\cdot s^{n})$
0	1	0.000902
0.1	0.9	0.0075
0.2	0.85	0.0252
0.3	0.73	0.15
0.5	0.6	0.67
0.7	0.49	2.75

**Table 2 micromachines-12-01494-t002:** Mesh independency test.

Mesh Elements	MI for Re = 80	MI for Re = 10
87,304	0.8924	0.9834
137,592	0.8994	0.9877
241,322	0.8996	0.9890
338,438	0.9017	0.9941
593,476	0.9017	0.9942

**Table 3 micromachines-12-01494-t003:** Comparison of current computational results for Pressure drop as a function of various Reynolds numbers.

Pressure Drop	Re	1	5	10	50	100
	Kurnia et al. [10]	1.48	7.5	15.22	86.59	205
	Present simulation	1.5	7.6	15.3	86.6	205.5

**Table 4 micromachines-12-01494-t004:** Position of the planes in the micromixer.

Plans	Z mm
P1	4.5
P2	6
P3	7.5
P4	9
P5	10.5
P6	12
P7	13.5
P8	16.5

## Data Availability

Data would be available upon reader’s request.

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
