# Peer review of "Mixing Enhancement of Non-Newtonian Shear-Thinning Fluid for a Kenics Micromixer"

_micromachines, 2021, doi:10.3390/mi12121494_

Round 1
Reviewer 1 Report
Line 68: good to mention the CFD code used. Was it a commercial or an in-house code ?
Line 104: Replace section title 'Mesh Test' to ' Grid independence test'
Line 11 3- 114: As no actual validation against experimental data was carried out, simply state: the microKenics mixer was benchmarked against three micromixers ...........
Reviewer 2 Report
In the manuscript, authors examine the mixing behavior of non-Newtonian shear-thinning fluids in a Kenics passive micromixer. The micromixer consists of two tubular inlet channels and a tubular mixing channel, in which six helical blades are positioned alternately to rotate fluid couples and develop so-called chaotic fluid flow. A numerical study is carried out using CFD simulations. Authors use the carboxymethyl cellulose (CMC) solutions with power-law indices among 1 to 0.49 and investigate fluid mixing for Reynolds numbers between 0.1 to 500. The content of the manuscript is of interest to micromachines readers.
I recommend major revisions for the reasons explained below,
- The research outcomes are not scientific unless the complete numerical method and fluid properties employed in the study are presented clearly. The manuscript gives some results, but these results cannot be reproduced by other researchers since the points below are not reported.
- What is the density of CMC solution(s)? Or how is it calculated? Please report.
- In the manuscript the diffusion coefficient (D) of CMC solution(s) is not reported. If this is on the order of 10-5-10-7 m2/s, only molecular diffusion is enough to yield very high MI at low Re numbers. So, readers will not know if microKenics design works well or not unless they know the magnitude of the diffusion constant. Please report the diffusion coefficient, used in the simulations.
- Which method (e.g., Finite Elements, Finite Volume, Finite Difference etc.) is used to solve the partial differential equations (PDEs) given in line 71-73? Please report.
- Which solver/software (e.g., COMSOL, ANSYS, Flow-3D, OpenFOAM etc.) is employed to solve governing equations? Please report.
- Which numerical schemes (1st order, 2nd order etc.) are selected to discretize the convection and diffusion terms in the PDEs in line 71-73. Please report.
- Since PDEs are solved, initial/boundary conditions must be reported. Without knowing these initial/boundary conditions, no one can reproduce the results in the manuscript. Please report the initial/boundary conditions determined.
- In equation 5, “D” is used to calculate Reynolds number (Reg). Is this the diffusion coefficient of the mixing tube diameter? Because only one “D” is defined in line 75. So, this may be confusing to readers and one may think that this is the diffusion coefficient of CMC. Please use a different symbol for tube diameter.
I would like to emphasize that without reporting/correcting the points in sections 1–6, the Figure 5, 6, and 7 in the manuscript are just Colorful Fine Display (CFD) instead of Computational Fluid Dynamics (CFD) simulations.
- Mesh study is not correct and numerical (false or artificial) diffusion in numerical solutions is completely disregarded. The literature review on numerical diffusion is missing. In the current literature, there are several studies that are dedicated to only the numerical simulations of advection dominant systems.
- In the manuscript, numerical diffusion is not mentioned even with one word. However, it is well known that in numerical solution of advection dominant systems numerical diffusion increases fluid mixing unphysically. Studies show that if tetrahedral elements are used in FVM, even simple flow systems (i.e., unidirectional flow where flow vectors are mostly orthogonal to the cell boundaries) and mild Re numbers produce significant amount of false diffusion. In Kenics passive micromixer, the flow pattern is constantly rotational in the mixing tube and you simulate up to Re 500. Therefore, if your numerical method is FVM, you report very erroneous and unphysical outcomes. In FEM, on the other hand, the type of stabilization method employed plays a significant role to control numerical diffusion. However, we do not know what is used in the manuscript. On top of that based on Figure 2 in the manuscript you employ quite coarse meshes and select a wrong parameter, which is vmax, in the mesh test. Since it is not reported what the vmax is, I assume this is the maximum velocity magnitude in the mixing tube. First, in numerical simulations of advection dominant flow systems, the total mesh density should be chosen large enough to be able to reveal numerical diffusion. The best method is to double mesh amount every time. In your case, you need to test a very fine mesh (e.g., 1-3 x 106) to ensure your system is mesh independent since you simulate up to Re 500. Second, in the mesh study MI must be selected as a test parameter instead of vmax because numerical diffusion occurs substantially during the numerical solution of convection diffusion equation (Eq. 3). If the vmax is used as a parameter to reveal grid related errors, the mesh test gives deceptive outcomes as it is in the manuscript. To present physical outcomes, please follow the steps below;
- Measure MI at the outlet of each mesh level
- Double the mesh size each time and use a very fine mesh to be able to select an optimum mesh level
- Use the highest Re number scenario in flow cases (Section 4 and Figure 2 shows a vmax, but we do not know what the Re scenario is and where this parameter is measured)
I strongly recommend reviewing the papers below to be able to report physical mixing outcomes;
https://doi.org/10.3390/pr7030121
https://doi.org/10.3390/mi9050210
https://doi.org/10.1146/annurev.fluid.29.1.123
https://doi.org/10.1115/1.2910291
https://doi.org/10.1016/j.ces.2011.02.036
- Presentation, organization, and punctuation problems.
- Line 22: please omit the word small because it is very subjective.
- Line 53: to evaluated: to evaluate
- Line 55-58 please divide this whole sentence to 2 or 3 sentences because the meaning is lost.
- Line 64-65: The purpose of the study is reported as “to investigate the performance of a microKenics for mixing shear thinning fluids, trying to have a high mixing quality and lower energy consumption and pressure drop.” However, the work in the manuscript does not serve to this purpose. Frankly, only one design is used in the manuscript and hence aiming to lower pressure drop and to increase mixing quality are not true. Alternative blade orientations could have been tested. Lowering energy consumption and pressure drop are the same thing, so please only state pressure drop since energy consumption is not computed and reported.
- Line 74-75: Please include the units of velocity, density, diffusivity etc.
- Line 93: Please use the citations appropriately in the sentence.
- Please use either Figure X. or Fig X. throughout the manuscript and captions.
- Please add XYZ axes to the Figure 1, 5, 6, and 7.
- Line 132: Please correct “vs. of”
- Line 138-139: Please correct sentence à cause the intense of fluid particles
- Line 143-145: Please correct the sentence because the meaning is completely lost.
- Line 145-150: Please correct the sentence and divide to 2-3 sentences because the sentence is too long to understand.
- Line 163: Please correct “for differences values”
- Line 163-168: please divide this whole sentence to 2 or 3 sentences because it is too long to understand.
- Line 170-174: Please check the meaning and divide this whole sentence to 2 or 3 sentences because it is too long to understand.
- Line 186-189: Please check the meaning and divide this whole sentence to 2 or 3 sentences because it is too long to understand
- Line 225-227: Please divide this whole sentence to 2 sentences.
- Line 237-241: Please divide this whole sentence to 2 sentences.
Reviewer 3 Report
This paper presents the numerical analysis of mixing enhancement of Non-Newtonian shear-thinning fluid for a Kenics Micromixer. The mesh independent test is performed and results show that the mixing performances of the Kenics micromixers is better than that of other micromixers in a low Reynolds number world. However, the novelty and numerical simulation details are not well presented. Due to these reasons, I recommend the publication of this paper with major revision.
The comments are as follows:
- The author may add one or two sentences in the introduction to highlight the novelty of the study.
- The author may add one plot to show the mesh of the numerical model.
- The author should add more details on how to perform the numerical model. For example, the boundary condition, the language or the software and convergence criteria for the numerical model.
- The author can add more information on how does this work contribute to other industrial applications.
- The author shows that Kenics Micromixer has better performance than other micromixers. The author should add more details on how does Kenics Micromixer affect flow field.
Round 2
Reviewer 3 Report
The authors' have answered and clarified all points raised well. I believe the additional clarifications to the methods and results make the paper much stronger. I now recommend the article for publication.